# New SARS-CoV-2 Infection Detected in an Italian Pet Cat by RT-qPCR from Deep Pharyngeal Swab

**DOI:** 10.3390/pathogens9090746

**Published:** 2020-09-11

**Authors:** Nicolò Musso, Angelita Costantino, Sebastiano La Spina, Alessandra Finocchiaro, Francesca Andronico, Stefano Stracquadanio, Luigi Liotta, Rosanna Visalli, Giovanni Emmanuele

**Affiliations:** 1Section of Microbiology, Department of Biomedical and Biotechnological Sciences, University of Catania, 95123 Catania, Italy; nmusso@unict.it; 2Department of Drug Sciences, University of Catania, 95125 Catania, Italy; angelita25costantino@gmail.com; 3Centro Veterinario Giarre, 95014 Giarre, Italy; selaspin1@me.com (S.L.S.); alessandra.fin1978@gmail.com (A.F.); 4Molecular Analysis and Biology Laboratory Biogene Catania, 95100 Catania, Italy; francesca.andronico@gmail.com (F.A.); biogene@biogene.it (R.V.); giovanniemmanuele@tiscali.it (G.E.); 5Department of Veterinary Science, University of Messina, 98122 Messina, Italy; luigi.liotta@unime.it

**Keywords:** SARS-CoV-2, pet cat, RNA-extraction, pneumonia

## Abstract

The pandemic respiratory disease COVID-19, caused by severe acute respiratory syndrome coronavirus 2 (SARS-CoV-2), emerged in Wuhan in December 2019 and then spread throughout the world; Italy was the most affected European country. Despite close pet–human contact, little is known about the predisposition of pets to SARS-CoV-2. Among these, felines are the most susceptible. In this study, a domestic cat with clear clinical signs of pneumonia, confirmed by Rx imaging, was found to be infected by SARS-CoV-2 using quantitative RT–qPCR from a nasal swab. This is the first Italian study responding to the request of the scientific community to focus attention on the possible role of pets as a viral reservoir. An important question remains unanswered: did the cat actually die due to SARS-CoV-2 infection?

## 1. Introduction

The World Health Organization (WHO) declared COVID-19 disease, caused by severe acute respiratory syndrome coronavirus 2 (SARS-CoV-2), as a worldwide pandemic [1]. The first epidemic cluster occurred in China, namely in Wuhan City [2,3], as early as 1 December 2019. As of mid-July 2020, there have been over 5,525,245 confirmed COVID-19 cases worldwide, with more than 30% cases in the EU and UK and more than 347,108 deaths globally [2], (https://www.worldometers.info/coronavirus, accessed on 10 May 2020). Italy was severely affected [4], and it was one of the first and hardest hit countries in Europe, with over 219,000 cases and 30,500 deaths reported [2], for which reason on 9 March 2020 a lockdown was declared for the entire country and progressively stricter restrictions adopted [4,5].

SARS-CoV-2 is structurally and functionally closely related to the already known coronaviruses responsible for the Middle East respiratory syndrome (MERS-CoV) and the severe acute respiratory syndrome (SARS-CoV). In detail, the mechanism of host cell infection is the same for SARS-CoV and SARS-CoV2, i.e., it depends on the external protein of the virus, the glycoprotein (S) spike, which is able to bind and recognize the human receptor angiotensin converting enzyme 2 (ACE2), primed by the transmembrane protease, serine 2 (TMPRSS2) and/or extracellular matrix metalloproteinase inducer CD147 [2,6,7,8].

Because of the close contact between people and pets, such as dogs and cats, the scientific community started to investigate the possibility of animal-human virus transmission [9]. Furthermore, the animal ACE2 receptor has high human ACE2 aminoacidic sequence identity [10,11,12]. It has been shown that some animal species, particularly felines, can occasionally become infected with SARS-CoV-2 [10,13,14,15,16]. So far, some cases of domestic animal infection have been reported in Belgium [17], Hong Kong [14], and France [18], but no cases had yet been investigated in Italy. In particular, viral RNA and infectious viral particles were found in the upper respiratory tract of domestic cats after introduction of SARS-CoV-2 virus samples through their nasal cavities; nevertheless, none of the infected cats showed clinical signs of the disease. In addition, viral RNA was detected in 1:3 healthy cats exposed to infected felines, suggesting that they had contracted the virus from the droplets exhaled by infected cats [13,19].

In this study, we identified the natural infection of a cat with SARS-CoV-2 for the first time in Italy. The RNA obtained from the nasal swab was processed by RT-qPCR using two different chemistries, revealing the presence of two SARS-CoV-2 genes. Finally, part of one gene was further sequenced to evaluate its nature.

## 2. Clinical Presentation

A sterilized, three-year-old, male European shorthair cat was presented to a veterinary clinic with the owner reporting serious respiratory distress of about three days. The mandatory vaccinations required by the current Italian legislation, including viral rhinotracheitis, calicivirus and panleukopenia, had been completed.

On physical examination, the cat had severe dyspnea and sialorrhea, Kussmaul breathing, and asynchronous chest and abdomen. Chest auscultation revealed increased vesicular murmur. Due to the presence of pathological infiltration and ground-glass opacity of the lungs, the differential diagnosis suggested an idiopathic interstitial pneumonia of bacterial, viral or mycotic origin. Despite amoxicillin and clavulanic acid administration in combination with aerosol, the cat died shortly after admission and the corpse was cremated according to the current health protocol.

## 3. Diagnostic Investigation

### 3.1. Materials and Methods

#### 3.1.1. Biochemical, Hematological and Imaging Analysis

According to good veterinary practice, a blood sample was collected from the jugular vein and 1 mL was transferred to a sterile tube containing ethylenediamine tetra-acetic acid (EDTA) for hematological analysis, whereas another aliquot of about 3 mL was collected in a sterile glass tube to obtain serum sample for biochemical analyses. Moreover, clinical imaging evaluation was performed using ultrasound and X-ray techniques.

#### 3.1.2. SARS-CoV-2 Detection and Molecular Analysis

Given the similar clinical picture and the current situation caused by SARS-CoV-2, a nasal swab was collected to test for SARS-CoV-2.

#### 3.1.3. Specimen Collection

A nasal swab was collected using the Virus Test Kit Diagnostics Sterile Pack Swabs Universal Viral Transport System (COD. RYCO-VART10B03, Jiangsu Rongye Technology Co., Ltd., Touqiao Town, Yangzhou City, China).

#### 3.1.4. RNA Extraction

To enhance RNA uptake, a variation to the standard protocol was made: 700 μL of swab buffer was processed in the same column, instead of the original 140 μL. AVE buffer and ethanol were added in the same proportions in order to reach a final extraction volume of 6.3 mL. Then, the total volume was eluted in the same column ten times. For the rest, the extraction process followed the protocol provided by the manufacturers.

Viral RNA from the original swab was purified using the QIAamp^®^ Viral RNA mini-kit (QIAGEN, Hilden, Germany; cod. 52904).

#### 3.1.5. RNA Quantification

The extracted RNA was quantified by fluorometric technique using the Qubit™️ RNA HS Assay Kit (Thermo Fisher, Waltham, MA, USA; cod. Q32852) according to the standard procedure.

#### 3.1.6. RNA Reverse Transcription

Nasal swab RNA reverse transcription was carried out using the QuantiTect^®^ Reverse Transcription kit (QIAGEN, Hilden, Germany; cod. 205311).

#### 3.1.7. Real-time RT-qPCR

RT-qPCRs targeting SARS-CoV-2 were performed using TaqMan and SYBR chemistries on a Rotor-Gene Q thermocycler (QIAGEN) to amplify two different SARS-CoV-2 genes: the N1 gene, as indicated by the Centers for Disease Control and Prevention (CDC), and the spike gene, respectively. The primers used, their concentrations and the qPCR thermal profiles are listed in Table 1.

Finally, amplicons were verified by running the RT-qPCR products on 1.8% agarose gel stained with (CANVAX, Córdoba, Spain) Greensafe DNA gel Stain (Cod. E0206) and FastRuler Ladder (Thermo Fisher, Waltham, MA, USA; cod. SM1103).

To exclude human mRNA cross-contamination, Human Beta-Actin Primers at a final concentration of 1 μM were used as a control at the SYBR RT-qPCR (Quantitech Primers, QIAGEN, Hilden, Germany; cod. QT00016786) giving an amplicon of 88 bp.

#### 3.1.8. PCR Amplification and Sequence Analysis

The amplicons obtained by PCR from the amplification with the N1 portion gene primers (without probe) were purified using the QIAquick PCR Purification Kit (QIAGEN, Hilden, Germany; cod. 28106) and quantified using the fluorimeter Qubit dsDNA BR Assay Kit (Invitrogen, Carlsbad, CA, USA; cod. 32850), then 5 ng of the product were sequenced in a SeqStudio Genetic Analyzer (Thermo Fisher Scientific, Waltham, MA, USA) using the Applied Biosystems BigDye terminator cycle sequencing 3.1v (Thermo Fisher Scientific, Waltham, MA, USA; cod. 4337455) as previously described [21]. Amplicons were then sequenced in a SeqStudio Genetic Analyzer (Thermo Fisher Scientific, Waltham, MA, USA) using the Applied Biosystems BigDye terminator cycle sequencing 3.1v (Cod. 4337455, Thermo Fisher Scientific, Waltham, MA, USA; cod. 4337455) as previously described [22] and compared with the reference sequence “MT077125 Severe acute respiratory syndrome coronavirus 2 isolated SARS-CoV-2/human/ITA/INMI1/2020 (complete genome sequence release date: 11-APR-2020)” using the BLAST tool (https://blast.ncbi.nlm.nih.gov/Blast.cgi).

#### 3.1.9. Ethical Disclosure

Ethical approval was not necessary as per institutional and national guidelines and regulations.

## 4. Results

### 4.1. Clinical Results

The main biochemical and hematological parameters revealed lower alkaline phosphatase and higher glycemia values, while the blood count test showed relative and absolute neutrophilia (Figure 1a).

The radiographic analysis revealed an unstructured interstitial pattern caused by the widespread presence of pathological infiltrate throughout the lung interstitium. The increased radiodensity of the lung parenchyma, defined as “ground glass”, with little or no evidence of the intrathoracic vessels, clearly confirmed this condition (Figure 1b). Furthermore, X-ray imaging revealed interstitial pneumonia with an area of pulmonary opacity (Figure 1b) leading to the suspicion of effusion, excluded by the ultrasound examination (data not shown).

### 4.2. Molecular Analysis Results

The RNA extracted from the nasal swab was quantified at 6.56 ng/μL and tested in duplicate by TaqMan Probes for the detection of the SARS-CoV-2 N1 Portion Gene. SARS-CoV-2 synthetic RNA and nasal swab RNA amplified within the 28th and 34th cycle threshold (CT), respectively, while no amplification for the No Template Control (NTC) was detected. The amplification curves and CT, average and SD data are reported in Appendix A (Appendix A). Furthermore, the real-time qPCR products were run on a 1.8% agarose gel to confirm the correct size of the amplicon (Appendix A, right).

Further evidence of SARS-CoV-2 was given by the detection of the spike gene within the 40th CT during the real-time qPCR assay, while no amplification for SARS-CoV-2 synthetic RNA and NTC was found (Appendix A). To ensure that two different real-time activities were conducted on the same RNA during the sample collection and processing phases, and to exclude any possible contamination with human RNA, a feline housekeeping gene (available in the Techne-FCoV Kit) was used to verify the presence of correct feline RNA. The nasal swab sample amplified within the 39th CT, while there was no amplification for the NTC (Appendix A). At the same time, the feline sample tested negative for the human beta-actin that was used as control.

Finally, the amplified fragment obtained by endpoint PCR using the same RT-qPCR primers (Appendix A, right) was sequenced using the Sanger method to verify the accuracy of the amplification, resulting in a perfect match with the SARS-CoV-2 N1 gene according to the BLAST tool (https://blast.ncbi.nlm.nih.gov/Blast.cgi) (Appendix A).

## 5. Discussion

In late December 2019, the severe acute respiratory syndrome-coronavirus 2 (SARS-CoV-2), identified as a novel coronavirus [23,24,25,26,27,28,29,30,31,32,33], first caused uncommon pneumonia in humans in Wuhan (China) and then rapidly spread internationally. Thus, the World Health Organization (WHO) named the disease caused by this virus “Coronavirus Disease 2019” (COVID-19) and officially declared COVID-19 as a pandemic [34].

There is evidence that SARS-CoV-2 shares 96.2% of its nucleotide identity with the RaTG13 coronavirus detected in horseshow bats in China [35]. Indeed, first evidence indicates that the infection took place through the consumption of meat derived from infected bats (bat meat is commonly eaten in China).

The fact that pets are in close contact is given, but it became especially relevant once it was shown that pets were also susceptible to infection. Knowing their susceptibility to SARS-CoV-2 is therefore very important as several cases of infected pets have been reported. Unlike several studies carried out on dogs, which have shown little, if any, susceptibility to the virus, felines—especially cats—seem to have greater predisposition to the infection [13,16].

In the same way as in other coronaviruses such as SARS-CoV, the feline coronaviruses described until now seem to be able to change their tissue tropism [36].

In this study, we report—for the first time in Italy—the case of a male European cat positive for SARS-CoV-2. Unfortunately, we were not able to confirm the presence of the virus in the lungs of the cat because it was cremated before any tissues could be collected. As such, no association can be made between detection of SARS-CoV-2 in the nasal swab of the cat and its clinical condition. SARS-CoV-2 may well have been the cause of the cat’s death, but it may have equally been an accidental finding [18].

The cat presented clinical signs of pulmonary disease and the blood test and subsequent X-ray and ultrasound investigation confirmed the diagnosis of severe pneumonia. As the cat’s pathology evolved rapidly and harmfully (the animal died in as little as three days), with clinical signs and rate of disease progression similar to human COVID-19 patients, and because previously published papers reported different cases of feline infection [10,13,14,15,16], a nasal swab was collected in order to verify a possible infection with SARS-CoV-2. The identification of SARS-CoV-2 virus was supported not only by the qPCR amplification of the commonly used N1 gene within the 34th CT, by means of the TaqMan probe, but also by the amplification, using a more economic SYBR green chemistry, of another portion of the SARS-CoV-2 virus spike gene—missing in the synthetic RNA commonly used as positive control—which may represent a new molecular identification target. At the same time, the feline sample was free from human cross-contamination, as confirmed by positive amplification for a feline housekeeping gene and by the absence of amplification of the human housekeeping gene beta-actin. Finally, to verify the presence of SARS-CoV-2, an endpoint PCR was performed with CDC primers targeting N1 and the products were subsequently sequenced using the Sanger method.

Although the clinical history and the investigations performed clearly show positivity with clinical manifestations for SARS-CoV-2 in the lung, the infection source could not be clarified. As a resident of a ground floor apartment, the cat was used to going outside. Moreover, it was the only pet living with the owner. Probably, the contamination was due to the habit of cats of licking potentially contaminated surfaces or through contact, not detected by the owners, with other unidentified positive cats or positive asymptomatic visitors. At the same time, the presence of SARS-CoV-2 in the nasal swab is not enough to assert that it caused the pathological event [37]. As per health protocols and due to the time required for the swab transport and analysis, the cat corpse was cremated before knowing the test results. For this reason, unfortunately it was not possible to perform histological analysis to confirm that the pet died due to SARS-CoV2, although the concomitant presence of severe pneumonia and virus RNA was ascertained. Therefore, the link between SARS-CoV-2 infection and the cat death remains an open question.

To date, the crisis that has affected several world regions seems to be over, but the natural reservoirs of the virus and its high contagion rate, as well as environmental and social conditions may pave the way for a second epidemic wave. Particularly, as cats and dogs are more common and in closer contact with humans than bats, they should be checked for SARS-CoV-2 when affected by severe pneumoniae, as several al studies have demonstrated the propensity of coronaviruses for interspecies transmission [38,39]. At this point, there is no evidence that cats can spread COVID-19 and owners should not abandon their pets nor compromise their wellbeing [18].

## Figures and Tables

**Figure 1 pathogens-09-00746-f001:**
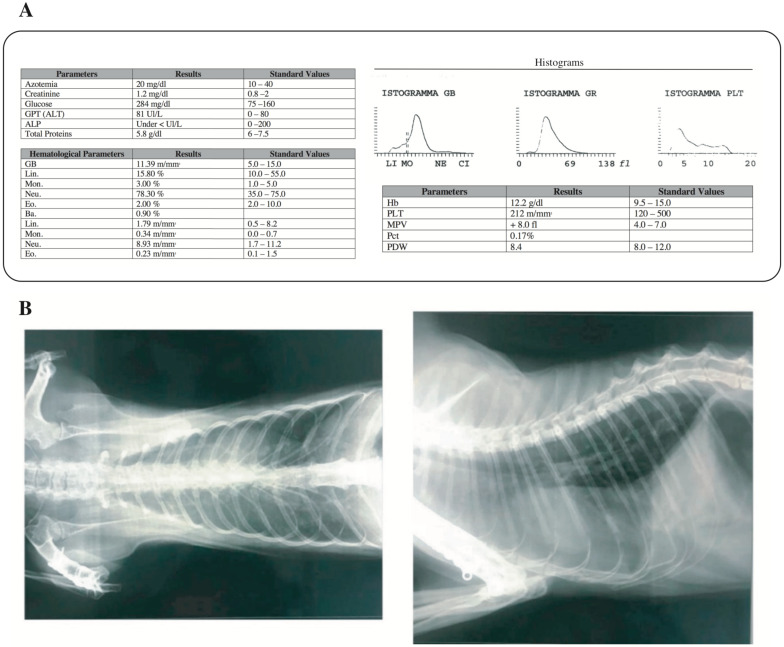
Clinical evaluation of the feline: biochemical and hematological parameters and X-ray imaging. (**a**) biochemical and hematological parameters of the collected blood sample. GPT (ALT): Glutamic Pyruvic Transaminase; ALP: Alkaline Phosphatase; GB: WBC (White Blood Cells); Lin.: Lymphocytes; Mon.: Monocytes; Neu.: Neutrophils; Eo.: Eosinophils; Ba.: Basophils; Hb: Hemoglobin; PLT: Platelets; MPV: Mean platelet volume; PDW: Platelet Distribution Width; Pct: Plateletcrit. (**b**) Lung X-ray diagnostic imaging. The opaque region of the image represents the ROI (Region of Interest) and reveals presence of pneumonia.

**Table 1 pathogens-09-00746-t001:** RT-qPCR specifications.

SARS-CoV-2 N1 Gene					
	Primer sequences 5′-3′	Conc.	Thermal Profile	Cycles	References
forward primer	5′-GAC CCC AAA ATC AGC GAA AT-3′	0.4 μM	95 °C for 5 s (denaturation)	50	[20,21]
reverse primer	5′-TCT GGT TAC TGC CAG TTG AAT CTG-3′	0.4 μM	60 °C for 5 s (annealing/extension and fluorescence data collection)		
probe	5′-FAM-ACC CCG CAT TAC GTT TGG TGG ACC-BHQ1-3′	0.2 μM			
SARS-CoV-2 Spike Gene					
	Primer Sequences 5′-3′	Conc.	Thermal Profile	Cycles	References
forward primer	5′-CGG CCT TAC TGT TTT GCC AC-3′	0.3 μM	95 °C for 15 min (PCR initial activation step)	50	Primers were designed on MT192773.1 Reference sequence by using the online software Primer3.
			94 °C for 15 s (denaturation);		
reverse primer	5′-TGT ACC CGC TAA CAG TGC AG-3′	0.3 μM	60 °C for 30 s (annealing)		
			72 °C for 30 s (extension and fluorescence data collection)		

RT-qPCR targeting SARS-CoV-2 was performed with TaqMan chemistry: the TaqMan^®^ probe (QIAGEN, Hilden, Germany) was labeled at the 5′-end with the reporter molecule 6-carboxyfluorescein (FAM) and at the 3′-end with the Black Hole Quencher 1 (BHQ-1) (Eurofins Genomics, Luxembourg); the reaction was performed using the QuantiNova Probe PCR (QIAGEN, Hilden, Germany; cod. 208252) according to the manufacturer’s recommendations. RT-qPCR targeting the SARS-CoV-2 spike was performed using SYBR RT-qPCR (Quantitech Primers, QIAGEN, Hilden, Germany; cod. QT00016786) giving an amplicon of 88 bp.

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
