# Peer review of "New SARS-CoV-2 Infection Detected in an Italian Pet Cat by RT-qPCR from Deep Pharyngeal Swab"

_pathogens, 2020, doi:10.3390/pathogens9090746_

Round 1
Reviewer 1 Report
This is an interesting paper on a topic of high international focus at the moment. Anything that increases our knowledge of the role, if any, of domestic pets in the spread or maintenance of this virus is of considerable interest.
The paper is generally well presented and the methods used in the study are clear.
Figure 1 requires a comprehensive, descriptive title.
There is, in my view, a weakness that could only be resolved with further work. In this cat there is good evidence of pneumonia and it is highly likely that this is what caused the fatal outcome. There is also good evidence that the coronavirus responsible for COVID 19 was identified in the nasal swab. There is however no solid evidence presented that the pneumonia is caused by or associated with the virus. What is really needed is for demonstration of the virus in the lung tissue by RT qPCR or preferably by immunocytochemistry in close association with the lesions in the lungs. Otherwise this is simply a report, still interesting, of detection of the virus in the nasal secretions of a cat that also happened to die of pneumonia.
If tissues are available to do this extra work that would be preferable. If not then the Discussion section should be amended to make it clear that there is no strong evidence here that the virus and the pneumonia are shown to be associated, let alone that there is a cause and effect. The title of the paper should also avoid suggesting any association.
Author Response
Q: Figure 1 requires a comprehensive, descriptive title.
R: done.
Q: There is, in my view, a weakness that could only be resolved with further work. In this cat there is good evidence of pneumonia and it is highly likely that this is what caused the fatal outcome. There is also good evidence that the coronavirus responsible for COVID 19 was identified in the nasal swab. There is however no solid evidence presented that the pneumonia is caused by or associated with the virus. What is really needed is for demonstration of the virus in the lung tissue by RT qPCR or preferably by immunocytochemistry in close association with the lesions in the lungs. Otherwise this is simply a report, still interesting, of detection of the virus in the nasal secretions of a cat that also happened to die of pneumonia. If tissues are available to do this extra work that would be preferable. If not then the Discussion section should be amended to make it clear that there is no strong evidence here that the virus and the pneumonia are shown to be associated, let alone that there is a cause and effect. The title of the paper should also avoid suggesting any association.
R: We agree with Reviewer 1 on the need to perform an histological analysis of the lung tissue of the cat. Unfortunately, due to the health protocols in force, as well as the time required for the transport and analysis of the swab, the cat's body was cremated before the test results were known. In detail, the logistics are organized as follows: the veterinary clinic, depending on the needs, books the collection of the test tubes containing the biological material to be analyzed and its transport by courier. The time that elapses between the collection of biological material and its arrival in the laboratory is approximately 24 hours; from that moment the acceptance and processing of the samples begins. During this time, the cat's body was cremated according to the health protocols. In this case, as soon as we understood the importance of the data, we tried in every way to recover even a simple slide of feline tissue, but it was not logistically possible. Consequently, it was not possible to perform histological analyzes to confirm that the animal died from SARS-CoV2. Conscious of this, we highlighted in the discussion that the correlation between pneumonia and SARS-CoV2 positivity remains unresolved. We added some information about these limitations in the discussion section (lines 156-161).
Q: The title of the paper should also avoid suggesting any association.
R: the title was modified as requested
Reviewer 2 Report
This manuscript describes a SARS-COV-2 natural infection in a pet cat. Though the authors claim that SARS-CoV-2 could be the cause of the death of the cat, the authors didn’t provide enough evidences for that. At least IHC should have been done to confirm the detection of the virus in the lung of the cat. Also, the author didn’t provide details about the histopathological findings detected in this case. Details about the macroscopic and the microscopic findings can help rule out other pathogens that could be the cause of the death. A variety of viruses, bacteria, fungi, and protozoa can cause respiratory infection in cat. Though, the author didn’t run any tests/bacterial culture to rule out these pathogens as the cause of the death. They only tested the sample for FCoV.
The author didn’t provide enough information about the cat such as age, vaccine history, whether it is the only cat/pet in the household or not. Was the cat kept indoor?
The authors need to add more information to the material and methods such as:
line 182 -187 the authors need to add the conc. of the primers and probe used, the thermocycler conditions used.
Line 194- Again what was the conc. of primers used? Thermocycler conditions used??
Line 199-What is the size of the amplicon?
Line 200- Conc. of the primers? The thermocycle conditions?
The references need to be revised. I wasn’t able to find ref. 37 and I don’t think that this ref. actually belongs to this paragraph (line 180-183). Also, Line 206- I don’t think that ref. 32 belongs to this paragraph .
There are some grammatical mistakes that need to be fixed for example
Line 42 change “peak” to spike
Line 192 “change according to the handbooks” to according to the manufacturer’s protocol.
Line 196 re-write this sentence
Sentences from 206-210 are a repeat of sentences 203-206.
Author Response
Q: Though the authors claim that SARS-CoV-2 could be the cause of the death of the cat, the authors didn’t provide enough evidences for that. At least IHC should have been done to confirm the detection of the virus in the lung of the cat. Also, the author didn’t provide details about the histopathological findings detected in this case. Details about the macroscopic and the microscopic findings can help rule out other pathogens that could be the cause of the death. A variety of viruses, bacteria, fungi, and protozoa can cause respiratory infection in cat. Though, the author didn’t run any tests/bacterial culture to rule out these pathogens as the cause of the death. They only tested the sample for FCoV.
R: We agree with Reviewer 2 on the need to perform an histological analysis of the lung tissue of the cat. Unfortunately, due to the health protocols in force, as well as the time required for the transport and analysis of the swab, the cat's body was cremated before the test results were known. In detail, the logistics are organized as follows: the veterinary clinic, depending on the needs, books the collection of the test tubes containing the biological material to be analyzed and its transport by courier. The time that elapses between the collection of biological material and its arrival in the laboratory is approximately 24 hours; from that moment the acceptance and processing of the samples begins. During this time, the cat's body was cremated according to the health protocols. In this case, as soon as we understood the importance of the data, we tried in every way to recover even a simple slide of feline tissue, but it was not logistically possible. Consequently, it was not possible to perform histological analyzes to confirm that the animal died from SARS-CoV2. Conscious of this, we highlighted in the discussion that the correlation between pneumonia and SARS-CoV2 positivity remains unresolved. We added some information about these limitations in the discussion section (lines 156-161). Unfortunately, this work was not planned as pure research as we received by the veterinary clinic only the pharyngeal swab and was not possible to obtain other clinical materials. Finally, we are not arguing that the cat died by SARS-Cov2 nor that pneumonia was caused by SARS-CoV2. Our data highlight only the presence of the virus in a cat with severe pneumonia. For these reasons we did not search any other pneumonia related microorganism. However, also the title was modified to avoid possible misunderstandings.
Q: The author didn’t provide enough information about the cat such as age, vaccine history, whether it is the only cat/pet in the household or not. Was the cat kept indoor?
R: we added these information in the veterinary clinical case and discussion sections (lines 63-66, 149-151). The cat was 3 years old, it lives in a ground floor apartment and used to go out, but it was the only pet living in the house. According to the veterinary, all the mandatory vaccinations were carried out (i.e. viral rhinotracheitis, calicivirus and panleukopenia).
Q: line 182 -187 the authors need to add the conc. of the primers and probe used, the thermocycler conditions used.
R: these information were added on lines 191-195 We chose to perform 50 cycles of amplification to avoid possible detection of nonspecific fluorescence emission.
Q: Line 194- Again what was the conc. of primers used? Thermocycler conditions used??
R: these information were added on lines 205-216. We apologize for missing information about SYBR chemistry, there was a typo.
Q: Line 199-What is the size of the amplicon?
R: added on line 216. The characteristics of human beta-actin gene were originally listed in the following table
Name |
Hs_ACTR1B_1_SG QuantiTect Primer Assay (QT00016786) |
Official Symbol |
ACTR1B [Human] |
Official Name |
ARP1 actin-related protein 1 homolog B, centractin beta (yeast) |
Species |
Human (Homo sapiens) |
Entrez Gene Id |
10120 |
Detected transcript(s) |
NM_005735 (2258 bp) |
Ensemble Transcript ID |
ENST00000289228 |
Amplified exons* |
1/2/3 |
Amplicon Length |
88 (NM_005735), |
Dye label / detection |
SYBR Green |
Suitable for two-step RT-PCR |
Yes, bioinformatically validated. |
Q: Line 200- Conc. of the primers? The thermocycle conditions?
R: information added on lines 215
Q: The references need to be revised. I wasn’t able to find ref. 37 and I don’t think that this ref. actually belongs to this paragraph (line 180-183). Also, Line 206- I don’t think that ref. 32 belongs to this paragraph .
R: the ref 37 is available online on the following link: https://pubmed.ncbi.nlm.nih.gov/32705153/ and the ref at line 206 was actually 38. These references were added to support the validity of the probe specificity and the sequencing protocol respectively.
Q: Line 42 change “peak” to spike
R: done, thank you (line 43)
Q: Line 192 “change according to the handbooks” to according to the manufacturer’s protocol
R: done, (line 203)
Q: Line 196 re-write this sentence
R: done, lines 207-208
Q: Sentences from 206-210 are a repeat of sentences 203-206.
R: We removed a sentence that actually was repeated. Moreover, the ref 32 at the end of this paragraph was a typo, the correct one is ref 38 (lines 223-225).
Reviewer 3 Report
In this communication authors report the detection of SARS-CoV-2 from a nasal swab of a cat with clinical signs of a pneumonia. The pneumonia was confirmed by Rx imaging. Based on these results authors suggest that the pneumonia might have been induced by the infection.
Major comment
As the authors mention an important question remains unanswered. Did the cat die from SARS-CoV2? However, another more specific question that has not been answered is if the pneumonia was caused by an infection with SARS-CoV-2. Unfortunately, the cat was not necropsied maybe because the owner did not allow this (reason is actually not mentioned). Obduction and detection of positive lung samples by PCR and immunohistochemistry would have increased the value of this case report.
A FCoV PCR was performed on the RNA extracted from a nasal swab to exclude the diagnosis of FIP. However, from this result a FIP infection cannot be excluded. First, the PCR that was used cannot distinguish between FCoV and more virulent FIPV strains. High levels of FCoV RNA in fluids or organs and especially detection of the mutations that correlate with FIP virus make a diagnosis of FIP more likely. In fact, FIP can only be definitely diagnosed by demonstration of FCoV antigen in lesions. More importantly no data are available showing that every cat with FIP will shed virus in nasal fluid. The source of the control sample of the other cat positive for FIP is not mentioned. But also if this was from a nasal swab the absence of FCoV RNA in the swab taken from the patient does not exclude FIP.
The spike gene specific amplification is proposed as a new molecular identification target. However, the spike gene is the most variable gene of the virus and therefore seems not the most obvious candidate for a diagnostic PCR. In this study the Ct value was also high suggesting low sensitivity. The synthetic RNA is not a proper positive control since it seems to lack the S gene sequences. At least the synthetic RNA sample was negative (fig S1 b). Without proper controls no conclusions can be drawn.
Other comments
- Young cats have been shown to be more susceptible for a SARS-COV-2 infection (ref Shi et al, 2020). Although obvious clinical signs were not reported (except one animal that died) massive lesions in nasal and trachea mucosa epithelium and lungs were found in the younger cats. The age of the cat could be mentioned (if known). Also, the histopathologic changes in SARS-CoV-2 experimentally infected cats should be presented and discussed in view of the pneumonia found in the cat form this study.
- Line 39. The wording of “”structure and function of the SARS-CoV2”” is not so clear. Do the authors mean structure and function of the virus? Or life cycle of the virus and pathogenesis?
- Line 42. ACE2 is a receptor for SARS but not MERS. Wording is not clear.
- Line 115-121. These sentences need to be rephrased. “” the question if SARS-CoV-2 could be transmitted to other animal species that could then become a reservoir of infection was not immediately a priority but became relevant as pets are in intimate contact with humans””. The fact that pets are in intimate contact is a given but it became especially relevant once it was shown that pets were also susceptible to infection.
- Line 122-124. A clear distinction should be made between tissue and host tropism. For FCoV generation of different virulent pathotypes leads to different tissue tropisms not cross-species transmission.
- Line 148. Contamination occurred through a third way. What is the first and second way?
- Line 150. “” No studies have yet been carried out that demonstrate a possible human-animal transmission””. On the other hand, there are several epidemiological observations clearly suggesting that cats might become infected through contact with a COVID-19 patient. This should be discussed in more detail.
- Line 156. Sentence: “” As cats- but also pets and farm animals… seem to be a natural reservoir for the virus””. This seems not correct. First cats are also pets and so far farm animals have not been shown to susceptible. Also, it is unknown if these animals can really become a real reservoir in which the virus is able to remain in the population and from which then new infections to humans can occur.
- Figure 2 is best taken up in the supplementary files.
Author Response
Q: As the authors mention an important question remains unanswered. Did the cat die from SARS-CoV2? However, another more specific question that has not been answered is if the pneumonia was caused by an infection with SARS-CoV-2. Unfortunately, the cat was not necropsied maybe because the owner did not allow this (reason is actually not mentioned). Obduction and detection of positive lung samples by PCR and immunohistochemistry would have increased the value of this case report.
R: We agree with Reviewer 3 on the needing to perform an histological analysis of the lung tissue of the cat. Unfortunately, due to the health protocols in force, as well as the time required for the transport and analysis of the swab, the cat's body was cremated before the test results were known. In detail, the logistics are organized as follows: the veterinary clinic, depending on the needs, books the collection of the test tubes containing the biological material to be analyzed and its transport by courier. The time that elapses between the collection of biological material and its arrival in the laboratory is approximately 24 hours; from that moment the acceptance and processing of the samples begins. During this time, the cat's body was cremated according to the health protocols. In this case, as soon as we understood the importance of the data, we tried in every way to recover even a simple slide of feline tissue, but it was not logistically possible. Consequently, it was not possible to perform histological analyzes to confirm that the animal died from SARS-CoV2. Conscious of this, we highlighted in the discussion that the correlation between pneumonia and SARS-CoV2 positivity remains unresolved. We added some information about these limitations in the discussion section (lines 156-161). Unfortunately, this work was not planned as pure research as we received by the veterinary clinic only the pharyngeal swab and was not possible to obtain other clinical materials. Finally, we are not arguing that the cat died by SARS-Cov2 nor that pneumonia was caused by SARS-CoV2. Our data highlight only the presence of the virus in a cat with severe pneumonia. For these reasons we did not search any other pneumonia related microorganism. However, also the title was modified to avoid possible misunderstandings.
Q: A FCoV PCR was performed on the RNA extracted from a nasal swab to exclude the diagnosis of FIP. However, from this result a FIP infection cannot be excluded. First, the PCR that was used cannot distinguish between FCoV and more virulent FIPV strains. High levels of FCoV RNA in fluids or organs and especially detection of the mutations that correlate with FIP virus make a diagnosis of FIP more likely. In fact, FIP can only be definitely diagnosed by demonstration of FCoV antigen in lesions. More importantly no data are available showing that every cat with FIP will shed virus in nasal fluid. The source of the control sample of the other cat positive for FIP is not mentioned. But also if this was from a nasal swab the absence of FCoV RNA in the swab taken from the patient does not exclude FIP.
R: As previously explained, we received only the pharyngeal swab. The veterinary clinic interest was only about knowing if the cat was positive to SARS-CoV2, not looking for the actual pneumonia-causing microorganism. According to the manufacture's handbook (HB10.01.08), the TECHNE qPCR feline coronavirus kit is able to detect all viral forms, even the most virulent ones. Our intent, looking for FIP, was only to exclude a nonspecific amplification of other coronaviruses (as a negative control) and to be sure that we were amplified only the human SARS-CoV2 virus RNA. Coronaviruses' N-gene portions are conserved among different species, in an early phase of pandemic by coronavirus is not possibile to exclude the presence of different coronavirus strains, but the positivity to human SARS-CoV2 was than ascertained by sequencing. We can not esclude a concomitant infection by FIP, even if our results highlighted the absence of this virus in the nasal swab. Finally, the amplification of FIP virus RNA performed on the FIP-positive cat serum and stool sample (the cat protidogram showed a 49% of gamma protein) was carried out only to be sure that the kit was correctly working.
Q: The spike gene specific amplification is proposed as a new molecular identification target. However, the spike gene is the most variable gene of the virus and therefore seems not the most obvious candidate for a diagnostic PCR. In this study the Ct value was also high suggesting low sensitivity. The synthetic RNA is not a proper positive control since it seems to lack the S gene sequences. At least the synthetic RNA sample was negative (fig S1 b). Without proper controls no conclusions can be drawn.
R: spike genes are variable, indeed we used the N-gene portions as suggested by CDC guidelines as a first screening. Spike gene portions were chosen to exclude possibile cross-contamination with the synthetic RNA in the kit containing only the N-gene portions and to improve the quality of our results showing the positivity of the sample to two different coronavirus gene portions using two different chemistries. We know that SYBR chemistry is less sensitive than taqMan chemistry, so the result is in line with our expectations about the cycle threshold. As we did not have positive control, the lack of amplification with synthetic RNA and NTC was an indirect confirmation of fluorescence specificity in SYBR.
Q: Young cats have been shown to be more susceptible for a SARS-COV-2 infection (ref Shi et al, 2020). Although obvious clinical signs were not reported (except one animal that died) massive lesions in nasal and trachea mucosa epithelium and lungs were found in the younger cats. The age of the cat could be mentioned (if known). Also, the histopathologic changes in SARS-CoV-2 experimentally infected cats should be presented and discussed in view of the pneumonia found in the cat form this study.
R: information about cat age and vaccinations were added at lines 63-66. Unfortunately, we did not perform any other analyses as the cat suddenly died and the corpse was cremated according to the health protocols.
Q: Line 39. The wording of “”structure and function of the SARS-CoV2”” is not so clear. Do the authors mean structure and function of the virus? Or life cycle of the virus and pathogenesis?
R: according to our knowledges, SARS-CoV2 is similar to MERS and SARS coronavirus both in terms of structure, function, life cycle and pathogenesis (Future Virol. 2020 May : 10.2217/fvl-2020-0050. Published online 2020 May 20. doi: 10.2217/fvl-2020-0050)
Q: Line 42. ACE2 is a receptor for SARS but not MERS. Wording is not clear.
R: We modified the sentence, line 42
Q: Line 115-121. These sentences need to be rephrased. “” the question if SARS-CoV-2 could be transmitted to other animal species that could then become a reservoir of infection was not immediately a priority but became relevant as pets are in intimate contact with humans””. The fact that pets are in intimate contact is a given but it became especially relevant once it was shown that pets were also susceptible to infection.
R: the sentence was modified according to your suggestions, thank you (lines 118-119)
Q: Line 122-124. A clear distinction should be made between tissue and host tropism. For FCoV generation of different virulent pathotypes leads to different tissue tropisms not cross-species transmission.
R: according to the reference, a change in the host tropism appears to be possibile for FCoV (Chang HW, Egberink HF, Halpin R, Spiro DJ, Rottier PJ. Spike protein fusion peptide and feline coronavirus virulence. Emerg Infect Dis. 2012;18(7):1089-95. doi:10.3201/eid1807.120143.)
Q: Line 148. Contamination occurred through a third way. What is the first and second way?
R: we apologize for the misunderstanding. We modified the sentence, now should be clearer (lines 149-153)
Q: Line 150. “” No studies have yet been carried out that demonstrate a possible human-animal transmission””. On the other hand, there are several epidemiological observations clearly suggesting that cats might become infected through contact with a COVID-19 patient. This should be discussed in more detail.
R: according to novel studies and our new information about the cat living habits, we agree with you about a possible, although remote, chance of human-cat cross-infection in this specific case. We modified the sentence (line 149-153)
Q: Line 156. Sentence: “” As cats- but also pets and farm animals… seem to be a natural reservoir for the virus””. This seems not correct. First cats are also pets and so far farm animals have not been shown to susceptible. Also, it is unknown if these animals can really become a real reservoir in which the virus is able to remain in the population and from which then new infections to humans can occur.
R: we modified the sentence according to your comments and new references 36 and 37 (lines 164-167).
Q: Figure 2 is best taken up in the supplementary files.
R: done
Round 2
Reviewer 3 Report
The authors addressed all the comments in their rebuttal and made several changes to the text based on the suggestions by the reviewers. To my opinion two important issues remain since this will lead to misinterpretation of the findings.
It is correct that the FCoV PCR kit that was used can detect all virulent types of FCoV including FIP strains. As such it cannot distinguish between the pathotypes. The reason to perform a FCoV PCR is valid and as authors mention in their rebuttal was only done to exclude a nonspecific amplification of other coronaviruses (as a negative control) and to be sure that only the human SARS-CoV2 virus RNA was amplified (which was subsequently also evidenced by sequencing). In this way it is also presented in the discussion: The negative result of real-time amplification of the feline sample with the Taqman probe for FCoV, excluded the possible positivity to a classic feline Coronavirus, responsible for FIP.” However the most important criticism is that one cannot conclude that FIP can be excluded based on a negative FCoV PCR of a pharyngeal sample. It does exclude the presence of FCoV in the sample ( so excludes possible nonspecific amplification with the SARS-CoV primers) but not the disease FIP. Actually authors seem to agree that FIP cannot be excluded as they mention in their rebuttal “”We cannot exclude a concomitant infection by FIP, even if our results highlighted the absence of this virus in the nasal swab”. This is in contrast with the firm statement in the last sentence of the introduction: “”In addition, we excluded the presence of feline infectious peritonitis (FIP), a lethal systemic disease often associated with feline coronavirus (FCoV).”
Second the study of Chang et al (ref 34) shows that a switch in virulence in FCoV strains can occur leading to a change in tissue tropism. You might consider different tissues as different hosts but this study does not claim (and is also not mentioned) that these mutations can lead to cross-species transmission. Therefore the suggestion is to remove cross-species transmission or add another reference supporting this claim.
Author Response
Q: Actually authors seem to agree that FIP cannot be excluded as they mention in their rebuttal “”We cannot exclude a concomitant infection by FIP, even if our results highlighted the absence of this virus in the nasal swab”. This is in contrast with the firm statement in the last sentence of the introduction: “”In addition, we excluded the presence of feline infectious peritonitis (FIP), a lethal systemic disease often associated with feline coronavirus (FCoV).”
R: we found your criticism very appropriate and modified the sentence according to your comment (lines 57-61): In this study we identified, for the first time, the natural infection of a cat by SARS-CoV-2 in Italy. The RNA extracted from the feline nasal swab was processed by RT-qPCR revealing the presence of two SARS-CoV-2 genes; a cross-contamination of the sample by the common feline coronavirus (FCoV) responsible for the feline infectious peritonitis (FIP) was excluded to ensure the specificity of our test. Finally, a part of the gene was further sequenced to evaluate its nature.
Q: Second the study of Chang et al (ref 34) shows that a switch in virulence in FCoV strains can occur leading to a change in tissue tropism. You might consider different tissues as different hosts but this study does not claim (and is also not mentioned) that these mutations can lead to cross-species transmission. Therefore the suggestion is to remove cross-species transmission or add another reference supporting this claim.
R: the sentence was shortened (lines 124-125): In the same way as other coronaviruses, such as SARS-CoV, the feline coronaviruses described until now seem to be able to switch their tissue tropism [34].
This manuscript is a resubmission of an earlier submission. The following is a list of the peer review reports and author responses from that submission.